# Observation of ballistic upstream modes at fractional quantum Hall edges of graphene

Ravi Kumar[1,9], Saurabh Kumar Srivastav [1,9], Christian Spånslätt [2,3,4], K. Watanabe [5], T. Taniguchi [5], Yuval Gefen[6], Alexander D. Mirlin[3,4,7,8] & Anindya Das [1✉]

The presence of "upstream" modes, moving against the direction of charge current flow in the fractional quantum Hall (FQH) phases, is critical for the emergence of renormalized modes with exotic quantum statistics. Detection of excess noise at the edge is a smoking gun for the presence of upstream modes. Here, we report noise measurements at the edges of FQH states realized in dual graphite-gated bilayer graphene devices. A noiseless dc current is injected at one of the edge contacts, and the noise generated at contacts at length, $L = 4\,\mu m$ and $10\,\mu m$ away along the upstream direction is studied. For integer and particle-like FQH states, no detectable noise is measured. By contrast, for "hole-conjugate" FQH states, we detect a strong noise proportional to the injected current, unambiguously proving the existence of upstream modes. The noise magnitude remains independent of length, which matches our theoretical analysis demonstrating the ballistic nature of upstream energy transport, quite distinct from the diffusive propagation reported earlier in GaAs-based systems.

[1] Department of Physics, Indian Institute of Science, Bangalore 560012, India. [2] Department of Microtechnology and Nanoscience (MC2), Chalmers University of Technology, S-412 96 Göteborg, Sweden. [3] Institute for Quantum Materials and Technologies, Karlsruhe Institute of Technology, 76021 Karlsruhe, Germany. [4] Institut für Theorie der Kondensierten Materie, Karlsruhe Institute of Technology, 76128 Karlsruhe, Germany. [5] National Institute of Material Science, 1-1 Namiki, Tsukuba 305-0044, Japan. [6] Department of Condensed Matter Physics, Weizmann Institute of Science, Rehovot 76100, Israel. [7] Petersburg Nuclear Physics Institute, 188300 St. Petersburg, Russia. [8] L. D. Landau Institute for Theoretical Physics RAS, 119334 Moscow, Russia. [9]These authors contributed equally: Ravi Kumar, Saurabh Kumar Srivastav. ✉email: anindya@iisc.ac.in

Transport in integer quantum Hall (QH) states occurs through one-dimensional edge modes located at the edge of the sample with downstream chirality dictated by the magnetic field (Fig. 1a). This is also true for particle-like fractional quantum Hall (FQH) states[1,2]. By contrast, so-called "hole-conjugate" FQH states ($i + 1/2 < \nu < i + 1$ with filling factor $\nu$ and $i = 0, 1, 2,..$) are expected to host counter-propagating chiral edge modes moving respectively along the downstream and upstream directions. A paradigmatic example is the $\nu = 2/3$ bulk state. MacDonald and Johnson[3,4] proposed that the edge supports two counter-propagating modes: a downstream mode, $\nu = 1$, and an upstream $\nu = 1/3$ mode (Fig. 1b-middle panel). The existence of upstream modes is of fundamental importance[5] and crucially affects transport properties including electrical and thermal transport[6,7], noise, and particle interferometry[8,9].

There has been an extensive effort over recent years to detect experimentally upstream modes and their properties in GaAs/AlGaAs quantum well based 2DEG. Two questions then come to mind. The first is whether the upstream modes can be detected by measuring the electrical conductance. Indeed, for the hole-conjugate $\nu = 2/3$ state, and for distances much smaller than the charge equilibration length, the two-terminal electric conductance $G$ is expected[10] to be $4/3 \frac{e^2}{h}$ (instead of $2/3 \frac{e^2}{h}$), confirming the counter-propagating character of edge modes. This value was indeed measured, but only on engineered edges at interfaces between two FQH states[11]. In experiments on a conventional edge (the boundary of a $\nu = 2/3$ FQH state), $G = 2/3 \frac{e^2}{h}$ is found to be robust and essentially equal to the filling factor $\nu = 2/3$, implying that charge propagates only in the downstream direction. This topological value of $2/3 \frac{e^2}{h} \equiv \nu \frac{e^2}{h}$ is expected universally in the regime of strong charge equilibration[10,12–14]; it does not tell anything about the presence or absence of upstream modes. For spin-unpolarized $\nu = 2/3$ state on a conventional edge in GaAs[15] an increase of $G$ from $G = 2/3 \frac{e^2}{h}$ to $\approx 0.73 \frac{e^2}{h}$ was detected at shortest lengths, thus providing an indication of counter-propagating modes. However, the conductance remained far from $4/3 \frac{e^2}{h}$. For a more conventional spin-polarized $\nu = 2/3$ state, also studied in Ref. [15], $G$ remained equal to $2/3 \frac{e^2}{h}$ down to the lowest distances. The second question is whether the upstream mode can be detected by measuring the thermal conductance. Thermal transport on an edge may be qualitatively different from charge transport[10,13]. Even if charge propagates only downstream, energy may propagate upstream. Measurements of thermal conductance $G_Q$ at $\nu = 2/3$ and related fillings yielded results fully consistent with the theory expected from hole-conjugated character of these states with upstream modes[16–19]. Still, measurements of $G_Q$ do not prove unambiguously the existence of upstream modes. Ideally, for $\nu = 2/3$, hole-conjugate state with counter-propagating 1 and 1/3 modes is favored by particle-hole symmetry, which is however only approximate in any realistic situation. Note, however, that an

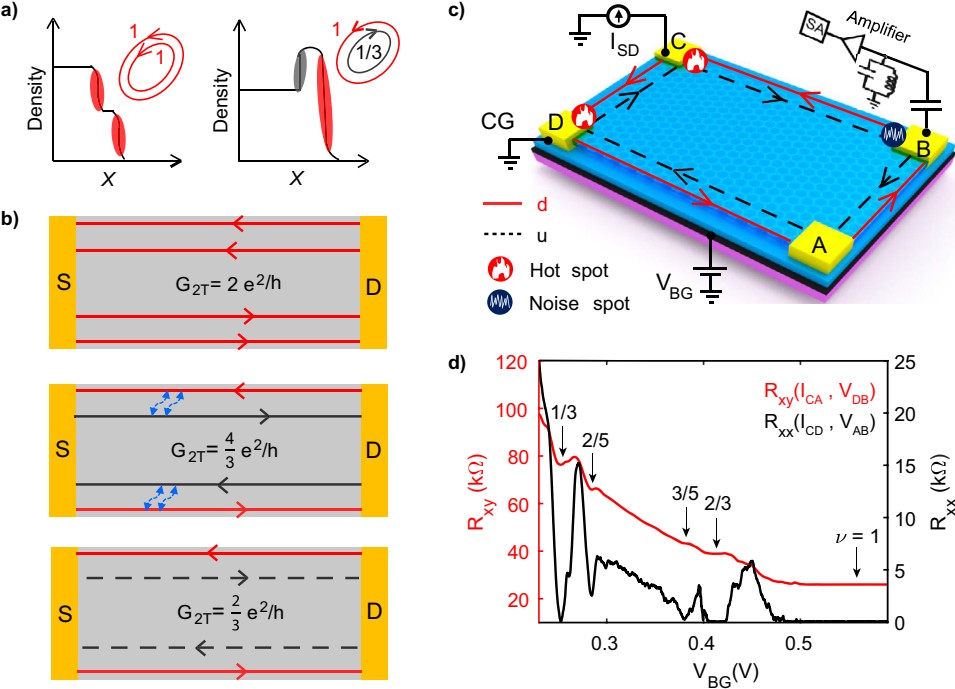

**Fig. 1 Edge profile, Device schematic, Hot spot, Noise spot, and QH response. a** Left panel: density profile at the edge of a sample with co-propagating mode structure for $\nu = 2$, giving rise to two-terminal conductance of $G_{2T} = 2e^2/h$ (cf. the top panel of (**b**)). Right panel of **a**: density profile at the edge of a sample with counter-propagating mode structure for $\nu = 2/3$, giving rise to the two-terminal conductance $G_{2T} = (4/3)e^2/h$ for sample length $L < l_{eq}^C$, where $l_{eq}^C$ is the charge equilibration length (cf. the middle panel of (**b**)). Here, $X$ is the coordinate across the edge. The bottom panel of **b** shows the structure of a $\nu = 2/3$ edge for $l_{eq}^C < L < l_{eq}^H$, where $l_{eq}^H$ is the thermal equilibration length, giving rise to $G_{2T} = (2/3)e^2/h$ with a downstream mode of conductance $(2/3)e^2/h$ and a charge-neutral upstream mode. The downstream charge modes are represented by the red solid lines, while the upstream charge mode is the black solid line and the charge-neutral upstream is the black dashed line in panel (**b**). The blue wiggly lines in the middle panel of **b** represent the interaction between the counter-propagating modes. **c** Schematics of the device and measurement setup, where the device is set into $\nu = 2/3$ FQH state. A noiseless dc current is injected at contact C and terminates into the cold ground (CG) contact D. The upstream mode carries the heat from the hot spot (where the energy is dissipated) near the contact C and reaches the noise spot close to contact B. The noise at this contact is then measured at a frequency of ~763 kHz using an LCR resonance circuit followed by amplifier chain and spectrum analyzer. **d** Transverse Hall resistance ($R_{xy}$) and longitudinal resistance ($R_{xx}$) of the device-1 at 10 T.

edge with two co-propagating $\nu = 1/3$ modes is a fully legitimate candidate[1,2] too. Such a model would be consistent not only with measured electrical conductance $G = 2/3\frac{e^2}{h}$ but also with the thermal conductance measurements reported in Ref. [18]. On a broader scope, several alternative approaches attempting to establish the presence of upstream current of heat at edges of a variety of FQH states in GaAs/AlGaAs structures were employed in Refs. [20–24]. Those studies used structures involving quantum point contacts or quantum dots. It is also worth noting that candidates to the non-Abelian $\nu = 5/2$ state possess a different number of upstream modes, and intensive current efforts aim at understanding which of them is actually realized in experiment[17,25–30].

The emergence of the two-dimensional graphene platform opened up a new era in the study of FQH physics[31,32], with different Landau level structure (as compared with traditional GaAs heterostructures), new fractions, and enriched family of quantum Hall states due to an interplay of spin, orbital, and valley degrees of freedom[33–38]. Furthermore, graphene features an unprecedented sharp confining potential and is thus expected to exhibit bulk-edge correspondence without additional complex edge reconstruction[39,40]. Also, in view of the sharp edge, one expects strong interaction between the edge modes in graphene, which may give access to regimes that are difficult to reach in GaAs structures. Moreover, the presence of the layer degree of freedom offers richer tunability in bilayer graphene[31,41]. Remarkably, for graphene or graphene-based hybrid structures, no direct evidence for the presence of upstream modes has so far been reported. Here, we report a smoking gun signature of upstream modes for hole-conjugate FQH states in bilayer graphene and identify their nature, employing noise spectroscopy, which is a purely electrical tool. The essence of our approach is as follows[42]. When a bias is applied to an FQH edge segment, the Joule heat is dissipated at the "hot spots" as shown in Fig. 1c. In the presence of upstream modes, heat is transported upstream to the so-called noise spot (Fig. 1c), where the heat partitions the charge current and thereby generates noise.

We have carried out electrical conductance together with noise measurements at integer QH states, electron-like state $\nu = 1/3$ and hole-conjugate states $\nu = 2/3$ and $\nu = 3/5$, realized in a dual graphite gated hexagonal-boron-nitride (hBN)-encapsulated high-mobility bilayer graphene (BLG) devices in a cryo-free dilution fridge at a base temperature of ~20 mK. For $\nu = 1/3$ state and integer QH states, we do not detect any excess noise along the upstream direction. This is expected because the corresponding edge states do not host upstream modes. By contrast, for $\nu = 2/3$ and $\nu = 3/5$ FQH states, a finite noise is detected which increases with increasing injected current. At the same time, the averaged current in the upstream direction is zero. Thus, noise detection unambiguously demonstrates that upstream modes exist for the hole-conjugate FQH states in graphene and only carry heat energy. Moreover, the magnitude of the noise remains constant for two different lengths $L = 10\,\mu m$ and $4\,\mu m$ between the current injecting contact and noise detection point in the same device. Moreover, our experimentally measured noise magnitude matches remarkably well with our theoretical analysis. This conclusively demonstrates the ballistic nature of upstream modes, implying the absence of thermal equilibration on the length scales employed in the experiment.

## Results

### Device and electrical response.
For our measurements, we have used two dual graphite gated hexagonal-boron-nitride (hBN)-encapsulated bilayer graphene devices, which are fabricated using the standard dry transfer pickup technique, see the Method

section and Supplementary Information (SI), section S1. To observe, at modest magnetic fields, well-developed FQH states, which are highly susceptible to disorder (below $\nu < 1$), screening layers are required. The presence of the top graphite gate serves that purpose and improves the quality of the device[43]. The device schematic is shown in Fig. 1c. The zero-magnetic-field gate response of the devices is shown in SI (S1). From this data, the mobility of the devices is extracted and found to be ~$1.7 \times 10^5 cm^2 V^{-1} s^{-1}$ and ~$3 \times 10^5 cm^2 V^{-1} s^{-1}$ for devices 1 and 2, respectively. First, we perform electrical conductance measurements at a fixed magnetic field of 10 T using standard lock-in amplifier. The electron density is tuned by the back graphite gate, keeping the top graphite gate fixed at zero voltage. The measured transverse Hall resistance ($R_{xy}$) and longitudinal resistance ($R_{xx}$) of device-1 are shown in Fig. 1d for $\nu \leq 1$. Clear plateaus in $R_{xy}$ are developed for $\nu = 1$, $\nu = 2/3$, and $\nu = 1/3$, accompanied with zeroes in $R_{xx}$ for $\nu = 1$ and $\nu = 2/3$, and a very low (in comparison to $h/e^2$) resistance of $30 - 40\,\Omega$ for $\nu = 1/3$. Similarly, plateaus are seen in $R_{xy}$ for $\nu = 1$, $\nu = 2/3$, $\nu = 3/5$ and $\nu = 1/3$ for device-2, see SI (S10), where $R_{xx}$ reaches zero for $\nu = 1$, $\nu = 2/3$ and $\nu = 1/3$, and ~$200\,\Omega$ for $\nu = 3/5$.

Next, we perform noise measurements at $\nu = 2/3$, $\nu = 1/3$, and integer QH states for two different lengths $L = 10\,\mu m$ and $4\,\mu m$ of device-1. The Measurement scheme for $L = 10\,\mu m$ is shown in Fig. 2a, where the device is set into $\nu = 2/3$ QH state with $d$ and $u$ representing counter-propagating downstream and upstream eigenmodes. The measurement scheme for shorter length $L = 4\,\mu m$ is obtained in the same device by changing the chirality (see SI, S5). Before the noise measurements, we perform two crucial checks: bias-dependent response of the $\nu = 2/3$ FQH state and chirality of charge transport. For this, a 100 pA AC signal is injected on top of the DC bias current at contact C, keeping contact D at the ground, and the AC voltage at contacts C and B is measured. The voltage at contact C, shown as $V_C$ in Fig. 2c, remains flat as a function of the DC bias current, which means the conductance of the $\nu = 2/3$ state does not change with the bias current. This kind of response, implying that there is no transport through the bulk (via states above the bulk gap), is a prerequisite check for the noise measurements. This is in full consistency with the value of the gap of $\nu = 2/3$ at 10 T, which is around 5 K as determined from the activation plot of $R_{xx}$ (see SI, S9), i.e., well above our largest bias voltage. Further, no detectable voltage is measured at contact B (shown as $V_B$ in Fig. 2c), which demonstrates that the charge propagates downstream only. Similar results are obtained for the shorter length (4 μm) (see SI, S5). Thus, if our $\nu = 2/3$ edge hosts counter-propagating modes (which is demonstrated below), the charge equilibration length is much smaller than the size of our device.

### Noise and analysis.
To unambiguously demonstrate the existence of an upstream mode, we injected a noiseless dc current at contact C and measured the voltage noise, $S_V$ at contact B along the upstream direction (Fig. 2a). For the noise measurement, an LCR resonant circuit with the resonance frequency of ~763 kHz is utilized together with an amplifier chain and a spectrum analyzer (see methods and SI(S4))[44–47]. At zero bias, the equilibrium voltage noise measured at the amplifier contact is given by[18,44]

$$S_V = G^2(4k_B TR + V_n^2 + i_n^2 R^2)BW , \qquad (1)$$

where $k_B$ is the Boltzmann constant, $T$ is the temperature, $R$ is the resistance of the QH state, $G$ is the gain of the amplifier chain, and $BW$ is the bandwidth. The first term, $4k_B TR$, corresponds to the thermal noise, and $V_n^2$ and $i_n^2$ are the intrinsic voltage and current noise of the amplifier. At finite bias, the upstream mode carries heat from the hot spot to the noise spot near the amplifier

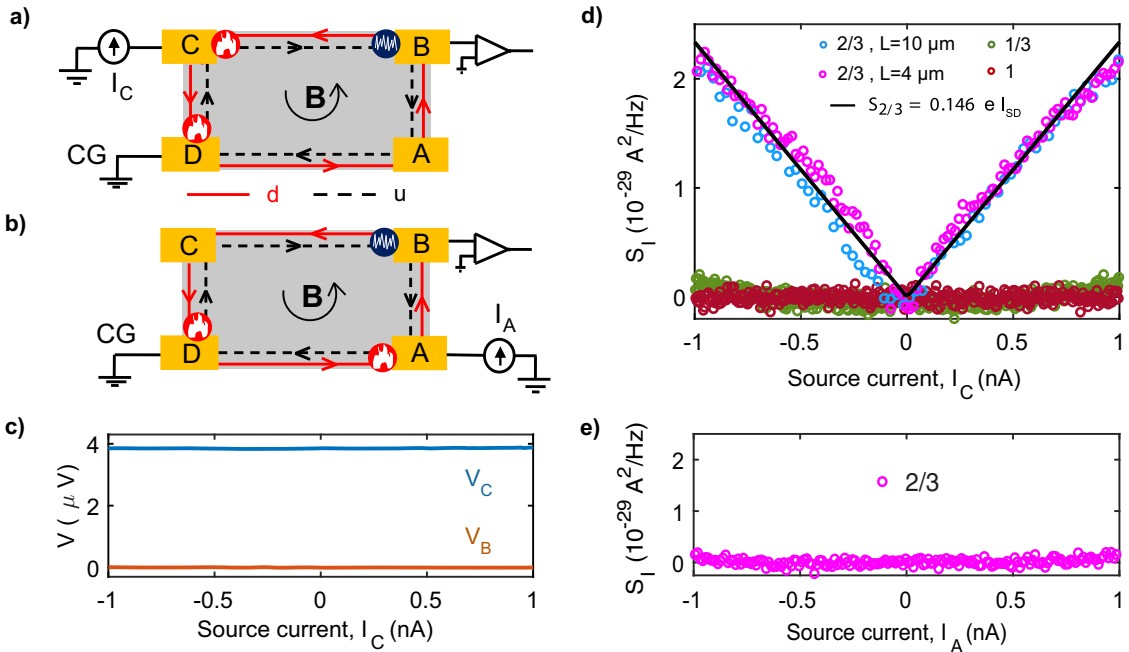

**Fig. 2 Noise data. a** and **b** show the noise detection scheme of the device-1 while the current is injected at contact C and contact A, respectively.
**c** Differential bias response of $\nu = 2/3$ state, where a 100 pA AC signal on top of the DC bias current is injected at contact C and the voltages are measured
at contact C ($V_C$) and B ($V_B$) using Lock-In amplifier. **d** Noise measured along the upstream direction for longer length (10 μm) as shown in Fig. 2a for $\nu = 1$
(brown circles), $\nu = 1/3$ (olive circles) and $\nu = 2/3$ (blue circles). The magenta open circles are the measured noise along the upstream direction for
shorter length (4 μm) by reversing the magnetic field (opposite chirality), and injecting current at contact A and measured noise at contact B (see SI, S5).
The solid black line is the theoretically calculated noise. **e** Noise measured along the downstream direction for $\nu = 2/3$ for the scheme shown in Fig. 2b.

contact. This leads to the enhancement of stochastic inter-mode tunneling processes in the noise-spot region, which creates an excess noise, see below for detail. At the same time, the intrinsic noise of the amplifier remains unchanged. Frequency independence of the thermal noise and the excess noise allows us to operate at a higher frequency (~763 kHz) so that we can eliminate the contribution from flicker noise (1/f) which usually becomes negligible for frequency above a few tens of Hz. The excess noise ($\delta S_V$) due to bias current is obtained by subtracting the noise value at zero bias from the noise at finite bias, i.e $\delta S_V = S_V(I) - S_V(I = 0)$. The excess voltage noise $\delta S_V$ is converted to excess current noise $S_I$ according to $S_I = \frac{\delta S_V}{R^2}$, where $R = \frac{h}{\nu e^2}$ is the resistance of the considered QH edge.

In the absence of an upstream mode, the energy cannot flow from the hot spot near C to the noise spot near B, see Fig. 2a, so that no noise is expected. This is precisely what is observed for $\nu = 1$ and $\nu = 1/3$ states, see Fig. 2d. At the same time, it is shown in Fig. 2d that for the $\nu = 2/3$ state there is a strong noise which increases almost linearly with current. This is quite striking as at contact B the time-averaged current is zero (Fig. 2c). This clearly demonstrates that the $\nu = 2/3$ edge hosts an upstream mode that leads to an upstream propagation of energy, even though the charge propagates downstream only. The mechanism of the noise generation is as follows[42]. The heat propagating upstream from the hot spot near C reaches the noise spot near B, inducing their creation of particle-hole pairs propagating in opposite directions. If the particle (or hole) is absorbed at contact B, while the hole (or, respectively, particle) flows downstream, there will be a voltage fluctuation at B detected by our noise measurement scheme. Similarly, the noise along the upstream direction is detected for $\nu = 2/3$ and $\nu = 3/5$ of device-2 as shown in SI(S10). Note that the 1/3 state gives rise to detectable noise at a larger bias current (Fig. 2d), which is, however, very weak: the noise magnitude remains ~15 times smaller than the noise measured

for 2/3 state. This tiny increase in noise for 1/3 as compared to $\nu = 1$ in Fig. 2d may be attributed to a minute amount of bulk contribution for 1/3 state. This is consistent with the bias response of the 1/3 state shown in SI(S12) where the conductance barely changes with the applied bias, thus confirming that the bulk contribution to transport in the 1/3 state is indeed tiny.

To verify that the heat propagates from the hot spot to the noise spot entirely via the edge, we have measured the noise in an alternative configuration. In this setting, a noiseless dc current is injected at contact A, while the contact C is electrically floating as shown in Fig. 2b. In this situation, in order to induce the noise, the heat would have to propagate upstream from the hot spot near D to the noise spot near B. However, this path is "cut" by the metallic contact C held at the base temperature. Thus, the only way for the heat to propagate is via the bulk. As shown in Fig. 2e, no detectable noise is measured at contact B, which rules out any sizeable bulk conduction of heat in our device.

To inspect the length dependence of the noise, we have also studied it for $L = 4$ μm of the same device-1 (see SI, S5). The data are shown in Fig. 2d. It can be seen that the noise amplitude is nearly identical for $L = 10$ μm and 4 μm. This is striking since it shows that the heat propagates upstream ballistically and without losses along the $\nu = 2/3$ edge, from the hot spot to the noise spot. There are two distinct mechanisms that could suppress the heat propagation and thus the noise: (i) thermal equilibration between the counter-propagating modes[10,13,42,48] and (ii) dissipation of energy from the edge to other degrees of freedom, including phonons, photons, and Coulomb-coupled localized states[49–51]. Our results show that none of these mechanisms is operative at bath temperature ($T_{\text{bath}}$) ~ 20 mK on the length scale of $L = 10$ μm. The absence of thermal equilibration on the edge is in a striking contrast with the very efficient electric equilibration emphasized above.

Up to now, all the data were at $T_{\text{bath}}$ ~ 20 mK. Now, we explore the effect of temperature. Figure 3a and b show the evolution of

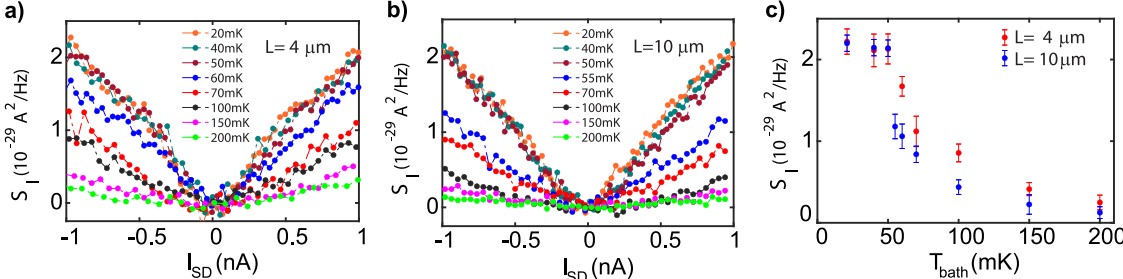

**Fig. 3 Temperature dependence of the noise. a** and **b** show the noise along the upstream direction at $\nu = 2/3$, at different bath temperatures for $L = 4\,\mu m$ and $L = 10\,\mu m$, respectively. **c** Noise value at $I_{SD} = 1\,nA$, extracted from the averaged slopes (positive and negative bias) of the linear fittings of the noise data (Fig. 3a and b), is plotted as a function of bath temperature for $4\,\mu m$ (red circles) and $10\,\mu m$ (blue circles) for $\nu = 2/3$. The error bar here is the standard deviation in averaged slope of the linear fit. The linear fittings of the noise data are shown in SI (S7).

the $\nu = 2/3$ noise at $L = 4\,\mu m$ and $L = 10\,\mu m$, respectively, with increasing $T_{bath}$. At each temperature, the noise data in Fig. 3a and b are fitted linearly for both positive and negative bias current, and from the averaged slopes we quote the noise value at $I_{SD} = \pm 1nA$ in Fig. 3c for both $L = 4\,\mu m$ and $L = 10\,\mu m$. The linear fittings are shown in SI (S7). It is seen that the noise remains constant and equal for both lengths up to $T_{bath} \sim 50\,mK$. We have verified [see SI (S6)] that the cooling of the electron system remains efficient down to our lowest temperature, i.e., the electron temperature $T_e$ is very close to $T_{bath}$ in the whole temperature range of our measurements. In particular, for $T_{bath}$ of 40 mK and 50 mK, the electron temperature $T_e$ is equal to $T_{bath}$. For the lowest temperature, $T_{bath} = 20\,mK$, we find, for the $\nu = 2/3$ state, $T_e = 22\,mK$, i.e., only slightly higher than $T_{bath}$.

For higher temperature, the noise decays with $T_{bath}$ (or, equivalently, with $T_e$), and the decay is substantially faster for the larger $L$. This decay can be attributed to one of two mechanisms mentioned above: thermal equilibration within the edge (which would imply a crossover from ballistic to diffusive regime of heat flow) or loss of heat to the bulk. Both mechanisms are expected to be enhanced at higher temperature[18,49–51], but further work is needed to understand which of them is dominant.

Our experiments thus clearly indicate that at low temperatures, $T_{bath} \leq 50\,mK$, the upstream heat transport is ballistic and lossless. To support this conclusion, we have calculated theoretically the expected noise $S_I$ on the $\nu = 2/3$ edge in this regime. The theory extends that of Refs. [42,48] to the ballistic (rather than diffusive) regime of heat transport corresponding to vanishing thermal equilibration. In this regime, the backscattering of heat takes place only at interfaces with the contact regions[10,19]. We assumed the bias voltage $V = (3h/2e^2)I_{SD}$ to be much larger than $T$, which is well fulfilled for our typical current $I_{SD} \sim 1nA$. The result (see method section and SI (S13) for detail)

$$S_I = 0.146\,eI_{SD} \qquad (2)$$

is shown in Fig. 2d and is an excellent agreement with the experimental data, thus giving a further strong support to our interpretation of the experiment.

## Discussion

Our measurements of noise present an unambiguous demonstration of the presence of an upstream mode in $\nu = 2/3$ and $\nu = 3/5$ FQH edges in graphene. This mode is responsible for the upstream heat transport that is at the heart of the noise generation mechanism. Remarkably, the noise is temperature-independent for $T_{bath} \leq 50\,mK$ and remains the same for $L = 4\,\mu m$ and $L = 10\,\mu m$, demonstrating the ballistic and lossless character of heat transport. The ballistic heat transport implies the absence of thermal equilibration on the edge, in contrast to full charge equilibration revealed by electric conductance

measurements. This is entirely consistent with the data of Ref. [18] on thermal conductance in graphene. There, a dramatic difference between the charge and heat equilibration lengths was explained by the vicinity of a system to a strong-interaction fixed point, where the bare modes of the $\nu = 2/3$ edge are renormalized into a charge and a neutral mode. Very recently, the absence of thermal equilibration (notwithstanding very efficient electric equilibration) was also reported for GaAs samples[19].

## Methods

**Device fabrication and measurement scheme.** For making encapsulated devices, we used the standard dry transfer pick-up technique[52,53]. Fabrication of these heterostructures involved mechanical exfoliation of hBN and graphite crystals on an oxidized silicon wafer using the widely used scotch-tape technique. BLG and graphite flakes were exfoliated from natural graphite crystals. Suitable flakes were identified under the optical microscope. The thickness of the top and bottom graphites were ~5 nm and ~20 nm, respectively, and the thickness of of the top and bottom hBN flakes were of the order of ~20 nm. The smaller distance between graphite gate and BLG layer (~20 nm thick bottom hBN) was a similar range to the magnetic length scale of our experiment at 10 T, implying a sharp confining potential at the physical edge of BLG. Details of the fabrication procedure are in SI (S1). The BLG channel area of the stack was microscopically ironed using an AFM (atomic force microscopy) tip in contact mode[54], to remove any atomic level strain or ripples or small bubbles from the channel area, which can arise due to the stacking process. After this, for making contacts we used electron beam lithography (EBL). After EBL, reactive ion etching (mixture of $CHF_3$ and $O_2$ gas) was used to define the edge contact. Then, thermal deposition of Cr/Pd/Au (4/12/60 nm) was done in an evaporator chamber having a base pressure of $\sim 1 - 2 \times 10^{-7}$ mbar. The optical image of the two measuring devices are shown in SI (S1). The schematic of the device and measurement setup are shown in Fig. 1c. All the measurements are done in a cryo-free dilution refrigerator having a base temperature of ~20mK. The electrical conductance was measured using the standard Lock-in technique. For $R_{xy}$, an ac current is injected at contact C (Fig. 1c), contact A is kept at the ground, and we measure the potential difference across the contacts D and B. For $R_{xx}$, we inject current at C, contact D is kept at ground, and we measure the potential difference across the contacts A and B. The resistance value of $R_{xy}$ for $\nu = 2/3$ is $\approx 39\,k\Omega = (3/2)h/e^2$, implying full charge equilibration among the counter-propagating eigenmodes. Under no charge equilibration, one would get a very different value of the resistance according to Landauer-Buttiker formalism, see SI (S11) for detail. The noise is measured employing noise thermometry based on an LCR resonant circuit at the resonance frequency of ~763 kHz and amplified by homemade preamplifier at 4K followed by room temperature amplifier, and finally measured by a spectrum analyzer. The details of the noise measurement technique are mentioned in the SI (S4).

**Theoretical calculation of noise.** We assume the regime of strong charge equilibration along the edge segment, i.e., $L \gg l_{eq}^C$, where $l_{eq}^C$ is the charge equilibration length. Then, the dc noise $S$ generated due to inter-mode tunneling along this segment is given by

$$S = \frac{2e^2}{h l_{eq}^C}\frac{\nu_-}{\nu_+}(\nu_+ - \nu_-)\int_0^L dx\,\Lambda(x)e^{-\frac{2x}{l_{eq}^C}}.$$

Here, $h$ is the Planck constant, and $\nu_+$ and $\nu_-$ are the total filling factors associated with the downstream and upstream edge modes, respectively, with the bulk filling factor $\nu = \nu_+ - \nu_-$. The exponential factor in the integral is a result of chiral charge transport, $\nu_+ \neq \nu_-$, and implies that the dominant noise contribution comes from

the noise spot—a region of size $\sim l_{eq}^C$ near the contact that is located on the upstream side of the segment (the voltage probe where the noise is measured).

The noise kernel $\Lambda(x)$ is calculated by assuming a thermally non-equilibrated regime. In this case, $\Lambda$ becomes $x$-independent within the edge segment, so that the $x$-integral is straightforwardly calculated, yielding for the $\nu = 2/3$ edge the noise

$$S_{2/3} = \frac{2e^2}{9h}\Lambda(V,\Delta) \, ,$$

where the dependence of $\Lambda$ on the bias voltage $V$ and the interaction between edge modes (parameter $\Delta$) is noted. We take $\Delta \approx 1$, which corresponds to the strong-interaction fixed point[5], at which the heat equilibration length is much larger than the charge equilibration length[18], as observed experimentally. To determine the dependence of $\Lambda$ on the voltage, we first calculate the effective temperatures of the downstream and upstream modes on the edge. They are found from the system of energy balance equations that include the Joule heating at the hot spot as well as a partial reflection of heat at interfaces between the interacting segment of the edge and the contact region. The contacts are modeled in terms of non-interacting one and 1/3 modes (see SI, S13). For $\Delta = 1$, the corresponding reflection coefficient is $\mathcal{R} = 1/3$. The resulting temperatures of the modes are

$$k_B T_+ = 0.13 eV \, , \qquad\qquad k_B T_- = 0.23 eV \, .$$

The noise kernel $\Lambda$ is now calculated by using the Green's function formalism for the chiral Luttinger liquid and the Keldysh technique. Expressing the result in terms of the bias current $I_{SD} = (2e^2/3h)V$, we come to the final result given by Eq. (2) of the manuscript. An analogous calculation for $\nu = 3/5$ edge yields $S_I = 0.138 \, eI_{SD}$, which is in a very good agreement with the experiment too (see SI, S13).

## Data availability

The data that support the findings of this study are available from the corresponding author upon reasonable request.

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

## Acknowledgements

C.S., Y.G., and A.D.M. acknowledge support by DFG Grants MI 658/10-1 and MI 658/10-2, and by the German-Israeli Foundation Grant No. I-1505-303.10/2019. Y.G. acknowledges support by the Helmholtz International Fellow Award, and by DFG RO 2247/11-1, and CRC 183 (project C01), and the Minerva Foundation. Y.G. further acknowledges the support of the Infosys Foundation. C.S. further acknowledges funding from the Excellence Initiative Nano at the Chalmers University of Technology and the 2D TECH VINNOVA competence Center (Ref. 2019-00068). K.W. and T.T. acknowledge support from the Elemental Strategy Initiative conducted by the MEXT, Japan and the CREST (JPMJCR15F3), JST. A.D. thanks the Department of Science and Technology (DST), India for financial support (DSTO-2051) and acknowledges the Swarnajayanti Fellowship of the DST/SJF/PSA-03/2018-19. A.D. further thanks for the Institute funding under the Institute of Eminence (IoE). R.K. and S.K.S acknowledge Inspire fellowship, DST and Prime Minister's Research Fellowship (PMRF), Ministry of Education (MOE), for financial support, respectively.

## Author contributions

R.K. and S.K.S. contributed to device fabrication, data acquisition, and analysis. A.D. contributed in conceiving the idea and designing the experiment, data interpretation, and analysis. K.W. and T.T. synthesized the hBN single crystals. C.S., A.D.M., and Y.G. contributed in the development of theory, data interpretation, and all the authors contributed in writing the manuscript.

## Competing interests

The authors declare no competing interests.
