## [Peer Review File · Nature Communications]

REVIEWER COMMENTS

Reviewer #1 (Remarks to the Author):

The manuscript reported an observation of ballistic upstream fractional quantum Hall modes in graphene. Different behaviors of noise vs source current have been found between filling factor $2/3$ and $1/3$. Similar experiments have been done for fractional quantum Hall effect and the authors successfully carried on the measurements in graphene devices. I have the following comments.

1. The plateau of $1/3$ isn't developed and R_{xx} only nearly reaches zero. How does S vs I_c look at a filling without FQHE? In addition, $1/3$ seems to show a slight slope in Fig2d when comparing with filling factor 1.

2. The noise signals are detected at $3/5$, where the bulk might not be fully insulating. What is the order of magnitude for the background noise from the bulk?

3. Figure 3a is puzzling. The noise results of 20mK and 40mK are hardly distinguishable, which might be explained by that the electron temperature is never cooled below 40mK. However, the data of 70mK and 100mK are also close to each other, and both far away from the 150mK, which is extremely puzzling.

4. I assume the Figure 3b's data are from filling factor $2/3$. The decaying behavior is expected, so the effective base temperature for this measurement is above 50mK? It might be interesting to check the ratio of 4 micron data to 10 micron data to see the length dependence above 50mK. I think the ratio is consistently below 1. Can the ballistic mode still be claimed?

5. In Figure S3, the lowest temperature in S3a is 20mK, and lowest temperature in S3b is slightly smaller than 0.2K. The other four points in S3b seem to be 1K, 775mK, 605mK and 475mK, respectively, same as those values in S3a. Are S3a and S3b from the same set of data?

Reviewer #2 (Remarks to the Author):

The manuscript from Ravi Kumar et al. reports on the experimental observation of upstream edge modes in fractional quantum Hall (FQH) states at the filling factors $\nu=2/3$ and $\nu=3/5$ in bilayer graphene. The main results are summarized below.

1. Excess upstream noise proportional to the injecting current was detected.
2. The average electric current in the upstream direction was measured to be nearly zero.
3. When the temperature is lower than 50mK, the amplitude of the detected upstream noise remains nearly identical for two different propagation lengths $L=4\mu\text{m}$ and $L=10\mu\text{m}$.

By combining the above results, the authors concluded that the edges of FQH states at $\nu=2/3$ and $\nu=3/5$ possess upstream modes that only carry energy but not charges. The existence of upstream neutral modes agree with the well-known picture of hole-conjugated FQH states. Furthermore, the upstream heat transport remains ballistic in the parameter range investigated in the experiment, which implies a lack of thermal equilibration in the sample. This feature is consistent with a recent experiment performed by the same group of authors (Ref. [15]).

I think the experiment was performed systematically and all procedures are explained clearly in the manuscript. The authors went through necessary checking to eliminate alternative explanations, which allowed them to reach the above conclusions unambiguously. Also, the experimental results agree nicely with the theoretical analysis provided in the manuscript. I can easily follow the presentation (both experimental and theoretical parts) in the manuscript. On the significance of the results, I agree with the authors that this work has provided more direct evidence for the existence of upstream neutral edge modes in $\nu=2/3$ and $\nu=3/5$ FQH states in bilayer graphene. This point is clearly addressed in the main text. Based on the clear presentation, significance of the results, and high quality of the work, I would recommend a publication of the manuscript in Nature Communications.

To further improve the manuscript, I list my suggestions below.

1. In the second paragraph of the main text, the authors cited Ref. [9] to discuss the expected two-terminal electrical conductance in non-equilibrated and equilibrated $\nu=2/3$ edges. I believe it is more respectful to cite Ref. [50] by Kane, Fisher, and Polchinski there as well.

2. In the same paragraph, the authors cited the two-terminal electrical conductance measurement in Ref. [10] to validate the existence of upstream modes at $\nu=2/3$. The article: "F. Lafont, A. Rosenblatt, M. Heiblum, and V. Umansky, *Science* **363**, 54 (2019)" also reported experimental signatures of a possible upstream electric current in the spin-unpolarized $\nu=2/3$ FQH edge in a short sample. Although the two-terminal conductance $4e^2/3h$ was not directly measured there, I think it may be appropriate to cite that article as well. As a follow-up question, can the authors briefly comment whether the $\nu=2/3$ FQH state in the bilayer graphene is expected to be spin-polarized or spin-unpolarized?

3. I think Fig. 1(b) can be more informative. For example, the figure can first illustrate a downstream charge mode responsible for $\sigma_{xy} = e^2/h$ and an upstream charge mode for $\sigma_{xy} = (1/3)e^2/h$. Then, the same figure shows that these edge modes after charge equilibration become a single downstream charge mode responsible for $\sigma_{xy} = (2/3)e^2/h$ and a neutral upstream mode. By doing so, the readers (especially non experts in quantum Hall physics) can better understand the importance of charge equilibration, and the underlying reason for the upstream mode carrying energy only. I suggest the authors include a corresponding figure for the $\nu=3/5$ FQH edge as well.

4. There are several typos in the Supplementary Material which I can notice. In S2, "If one plot" should be If one plots. In S5, "remains same" should be remains the same. In S8, there is a typo " $G_{j \leftarrow i}$ ".

Reviewer #3 (Remarks to the Author):

This work studies noise measurement in bilayer graphene in the FQHE regime. The authors observed current-dependent excess noise in filling factors $2/3$ and $3/5$ but not in $1/3$ and 1 . The observations provide evidence of upstream modes in the hole-conjugate states. The manuscript is concise and clearly written.

I have a few questions/comments.

1) In the abstract the authors described the devices studied in this work to be "dual-gated bilayer graphene". But there is no description in the manuscript or the supplementary information

regarding the dual-gate, neither in terms of device fabrication nor measurement. What is the significance of dual-gate here? And is the 2DES being bilayer graphene important? What do the authors expect for monolayer graphene?

2) From all what I can see in the manuscript and in the supplementary information, the noise measured here appears to be a voltage noise. For example, Figure S3 which plots S_V is easy to understand. Under a noiseless current, there are voltage fluctuations while the current should conserve. But in the main text the authors plot S_I . How did the authors convert to the current noise?

3) There are various sources of noise (thermal noise, $1/f$ noise, shot noise, etc). Since noise measurement is the technical key of this work, the authors should have a more in-depth discussion on the noise both in terms of measurements and analysis. In the current manuscript this part is rather too vague.

4) It helps to include some basic characteristics of the devices, either in the main text or in the supplementary information. For example the author claim "high mobility" in the main text. What is the mobility here?

We hereby resubmit the manuscript NCOMMS-21-29266.

Please find below our detailed response to the Reviewer's comments / questions / recommendations. While responding to each of the comments, we also specify the changes made in the Manuscript and the Supplementary Material as a reaction to this comment.

Reviewer # 1 (Remarks to the author):

'The manuscript reported an observation of ballistic upstream fractional quantum Hall modes in graphene. Different behaviors of noise vs source current have been found between filling factor $2/3$ and $1/3$. Similar experiments have been done for fractional quantum Hall effect and the authors successfully carried on the measurements in graphene devices. I have the following comments.'

We are delighted to read that the Reviewer deems our experiments on graphene devices successful. We agree with the Reviewer that similar experiments have been performed on fractional quantum Hall (FQH) edges in GaAs/AlGaAs based two-dimensional electron gas (2DEG) system. We would like to emphasize, however, that those studies point towards a diffusive nature of heat transport in the $\nu=2/3$ FQH state, as expected at length scales above the thermal equilibration length. Our result on graphene demonstrates for the first time the ballistic nature of upstream heat flow in the $\nu=2/3$ state, an observation which has not been made until the present and a parallel work (**arXiv preprint arXiv:2106.12486 (2021)**). The latter has demonstrated the presence of ballistic upstream heat flow in GaAs system. Notably, we have also an excellent quantitative agreement between theory and experiment for the noise amplitude. In short, our work not only pioneers such noise studies in graphene, but also goes well beyond previously published experiments on GaAs/AlGaAs structures.

We further thank the Reviewer for her/his useful questions, comments, and recommendations. We fully agree with the Reviewer that it was important to further improve the manuscript by clarifying those points for the readers. In the new version of the manuscript, we have taken into account all the comments by the Reviewer. Below we address them one by one and specify the changes made in reaction to each of them.

'1. The plateau of 1/3 isn't developed and R_{xx} only nearly reaches zero. How does S vs I_c look at a filling without FQHE? In addition, 1/3 seems to show a slight slope in Fig2d when comparing with filling factor 1.'

We agree with the Reviewer that the 1/3 plateau is not as robust as the 2/3 plateau, as can be seen in Fig. 1d of the manuscript. The minima of R_{xx} at the 1/3 plateau reach approximately 30-40 Ohm (which is a nearly zero value on the scale of characteristic quantum Hall resistances). As pointed out by the Reviewer, the 1/3 noise data (Fig. 2d in the manuscript) exhibit a small slope yielding a detectable noise at the 1 nA bias current. However, the noise amplitude remains ~ 15 times smaller compared to the noise measured for 2/3 state at $I_{sd} = 1$ nA. This tiny increase in noise for 1/3 as compared to $\nu=1$ can be attributed to a minute amount of bulk contribution for the 1/3 state.

As a reaction to this comment by the Reviewer, we have added in the revised manuscript (section "Noise and analysis") a discussion of the residual noise on the 1/3 plateau. Furthermore, in order to provide an additional verification of the smallness of the bulk contribution in the 1/3 state, we have added data on the bias response (revised Supplementary Material [SM], Section S12). The corresponding measurement scheme is now shown in Fig. 13a of the SM; for convenience, it is also reproduced here – see panel (a) of the figure below:

The measurement scheme involves a 100pA AC signal from the lock-in, superimposed on a DC bias current. It is injected at contact C, and the AC voltage at contact C and contact B is measured. The voltage at contact C (shown in red) and contact B (shown in blue) are shown in panel (b). The voltage at contact C remains almost flat, which confirms that the conductance of the 1/3 state barely changes with applied bias. The nearly zero voltage signal at contact B demonstrates that current flows almost entirely in the downstream direction. These observations confirm that contribution of bulk transport in the $\nu=1/3$ state is negligibly small. This is in full consistency

with a very small value of the noise measured in the upstream direction at $\nu=1/3$, as shown in Fig. 2d of the manuscript. This is also consistent with the measured activation gap of $1/3$, which is $\sim 200\mu\text{eV}$ [see revised SM, Section S9] and thus substantially higher than the applied bias range ($\sim 80\mu\text{eV}$) in our experiment. Finally, it is worth mentioning that the $1/3$ state for device-2 exhibits a good QH plateau (Fig. S11a in the revised SM) with $R_{xx} \sim 0$. As a result, we do not detect any noise that would be induced by the applied bias (Fig. S11b in the revised SM, olive open circles).

In relation to the $1/3$ state, the Reviewer asks also about the behavior of the noise without FQHE. We understand that the Referee wants to check whether there is a clear difference between the behavior of noise at the $1/3$ “nearly plateau” of device-1 and at a non-FQHE fraction (such as e.g. $\nu=1/2$ state). Such a difference is, of course, expected: As we have demonstrated, the $1/3$ state exhibits a very low (even if not exactly zero) bulk conduction, whereas the $1/2$ state is a good bulk conductor. We have verified that this is indeed the case, and present this in the following figure:

The figure displays the noise measured away from a FQH filling, at $\nu=1/2$. A strong noise is observed, reflecting a good bulk transport ($R_{xx} \sim 5 \text{ KOhm}$ in Fig. 1d of the manuscript). This value is approximately 40 times larger than a very weak noise at $\nu=1/3$, thus providing a further confirmation of the fact that the bulk contribution for $\nu=1/3$ is negligibly small in our experiment.

‘2. The noise signals are detected at $3/5$, where the bulk might not be fully insulating. What is the order of magnitude for the background noise from the bulk?’

The background noise from the bulk at $\nu=3/5$ state is shown in Fig. S11d in the revised SM. As pointed out by the Reviewer, a weak noise was detected. As explained in the figure caption to Fig. S11, we attribute it to a contribution of heat transport through the bulk, possibly via the mechanism identified in Ref. [Phys. Rev. B 93, 245427 (2016)]. The fact that this background noise is larger in the $3/5$ state than in the $2/3$ state is quite natural, in view of a smaller value of the gap in the $3/5$ state. On the other hand, this background noise remains weak also in the $3/5$ state: comparison of panels d and c of Fig. S11 of the revised SM shows that the background noise

is ~ 8 times smaller than the noise measured along the upstream direction (i.e. the one due to upstream edge heat transport) in the same $3/5$ state.

'3. Figure 3a is puzzling. The noise results of 20mK and 40mK are hardly distinguishable, which might be explained by that the electron temperature is never cooled below 40mK. However, the data of 70mK and 100mK are also close to each other, and both far away from the 150mK, which is extremely puzzling.'

We thank the reviewer for raising this important question about the electron temperature, which was not described in sufficient detail in the original version of the manuscript. In our previous works [**Physical Review Letters 126, 216803 (2021)** and **Science advances 5, eaaw5798 (2019)**], we have shown how the electron temperature was precisely determined. For our cryo-free dilution fridge we have achieved the base electron temperature in the range of ~ 20 -30mK.

In response to this question by the Reviewer, we have performed a careful analysis of the electron temperature in the context of the present experiment. The results are presented in the section "Noise and analysis" of the revised manuscript and in the newly added section S6 'Gain and electron temperature estimation' in the revised SM. We also summarize them here. Most importantly, we find that the cooling of the electron system remains very efficient: the electron temperature remains very close to the bath temperature in the whole temperature range. In particular, we find that for the lowest bath temperature (T_{bath}) of 20 mK, the electron temperature (T_e) on the $2/3$ plateau is only slightly higher: 22 mK.

As shown below figure (left panel), in our measurement setup, we have used extra low-pass filter made of 200 ohm and 1 nF capacitor for each line and each line is thermally anchored to

the cold finger using silver paint. This helps to get low electron temperature down to approximately 20 mK in our experiment. The bath temperature was measured using the calibrated RuO₂ sensor sitting at the mixing chamber plate. For calculating the electron temperature, we measure the thermal noise at zero bias at a given filling factor e.g. $\nu=1$,

$$S_v = G^2(4k_B T R + V_n^2 + i_n^2 R^2) B W$$

Here $4k_B T R$ is thermal noise, and V_n^2 and i_n^2 are intrinsic voltage and current noise of the amplifier respectively. R is resistance of the device at given filling factor, T is the bath temperature (temperature of mixing chamber (MC) plate), and G and BW are the gain and bandwidth of the amplifier, respectively. At an integer quantum Hall plateau, any change in bath temperature will only affect the first term in above equation, while all other terms are independent of temperature. First, we calculate the gain G by measuring S_v at different bath temperatures and then by plotting S_v/BW versus temperature (in the range of 175mK – 1K), the slope of the linear fit gives $G^2(4k_B R)$, as shown in the middle panel of the figure depicted above (solid red line). Since the resistance is exactly known for $\nu=1$, we infer the gain G to be ~ 1250 . Once the gain is known, one can calculate the $(V_n^2 + i_n^2 R^2)$ from the intercept of the S_v/BW versus temperature plot. It should be noted that we do not use the base temperature data (20 mK) for the linear fitting for the gain and the amplifier noise determination, because at base temperature the electron temperature might deviate from the bath temperature. However, at higher temperature such as 50mK or above the bath temperature and electron temperature are expected to be same, and hence it will give minimum error in calculating the gain and intrinsic noise of the amplifier. The dashed line in the above figure is the linear extrapolation.

Once the gain (from the slope) and intrinsic noise of the amplifier (from the intercept) are determined, so now from the known value of the measured noise at a bath temperature (shown by the star in the above figure), the corresponding electron temperature can be found directly using the following equation:

$$T_e = \frac{\left(\frac{S_v}{G^2 BW}\right) - (V_n^2 + i_n^2 R^2)}{4k_B R}$$

For the $\nu = 1$ plateau, our measured value of noise at base temperature of 20mK is $1.1 \times 10^{-8} \text{ V}^2$, corresponds to $T_e \sim 27\text{mK}$, which is consistent with our previous works (**Physical Review Letters 126, 216803 (2021), Science advances 5, eaaw5798 (2019)**). Furthermore, as shown in the right panel of the figure above, we have carried out a similar analysis to determine the electron temperature from the noise data of $\nu=2/3$ QH plateau. In this case, the electron temperature is found to be 22mK for the base temperature (bath temperature 20 mK). It can be seen in the figure that the noise data below 60mK fall on the linear extrapolation, thus confirming that the electron temperature in our setup is very close to the bath temperature in the working temperature range.

We have thus demonstrated---and explained in detail in the revised manuscript (in section “Noise and analysis”) and in the newly added section S6 ‘Gain and electron temperature estimation’ of the revised SM---that the electron temperature is essentially equal to the bath temperature in our experiment, and in particular in Fig. 3. Therefore, the low-temperature ($T < 50 \text{ mK}$) saturation of noise in Fig. 3 is not a consequence of a poor cooling of the electron system but rather an intrinsic property of the temperature dependence of the noise. At low temperatures T , the thermal equilibration length becomes larger than the system size, and the noise saturates at the

value corresponding to its zero-T limit. The limiting value is L-independent due to the ballistic character of the upstream heat transport. The measured low-temperature value of the noise is also in an excellent agreement with our theoretical calculations (assuming no heat equilibration, e.g., the low-temperature regime), thus providing a further confirmation of our interpretation of the experiment. Thus, the observed L-independent saturation of noise magnitude at T below 50mK clearly establishes the ballistic nature of the upstream heat flow at $\nu=2/3$.

The Reviewer also asks about the noise data points at higher temperatures (70 mK, 100 mK, and 150 mK). In particular, the Reviewer mentions that the values for 70mK and 100mK are relatively close. In fact, there is a clear difference (of order of 30%) between them for L=4 μ m (Fig. 3a); it becomes larger (roughly factor of 2) for L=10 μ m (Fig. 3b). Motivated by this question of the Reviewer, in the revised version of the manuscript we have reanalyzed the noise data at each temperature by linear fitting for both the positive and negative bias current. This is presented in the newly added section S7 “Noise analysis” in the revised SM. There, each panel presents a fitting at a given T and given sign of the current and the resulting value of the slope yielding the noise value at $I_{SD}=\pm 1$ nA. The resulting values are averaged over the sign I_{SD} and plotted as a function of the bath temperature in Fig. 3c of the revised manuscript. There, one observes a decay of the noise with T at elevated temperatures as expected. As discussed in the paper, this decay is due to one of two mechanisms: equilibration within the edge or dissipation of energy to other degrees of freedom. However, the data that we have, with substantial error bars, are not sufficient to make unambiguous conclusions concerning the form of the T dependence (and thus the underlying mechanism). As mentioned in the paper, this is left for future research. Our main focus in this work is on the low-temperature data.

‘4. I assume the Figure 3b’s data are from filling factor 2/3. The decaying behavior is expected, so the effective base temperature for this measurement is above 50mK? It might be interesting to check the ratio of 4-micron data to 10 micron data to see the length dependence above 50mK. I think the ratio is consistently below 1. Can the ballistic mode still be claimed?’

Figure 3b (now Fig.3c in the revised version) indeed corresponds to $\nu=2/3$. We have explicitly pointed this out in the figure caption in the revised version. Furthermore, we have added in this figure the data for the current dependence of the noise for L=10 μ m (Fig. 3b of the revised version). In addition, we have added a section S7 to the SM, which describes how the analysis of the data in Figs. 3a and 3b (which results in Fig. 3c) is performed. We have also added the corresponding clarifications in the caption of Fig. 3.

The question of the Reviewer concerning the T dependence partly overlaps with the Reviewer’s previous question (item 3). As detailed in our response there, the electron temperature in our experiment is essentially equal to the bath temperature. In particular, for the lowest bath temperature of 20 mK, the electron temperature on the 2/3 plateau is found to be 22 mK. The corresponding analysis is presented in Section “Noise and analysis” in the revised manuscript and in the newly added section S6 ‘Gain and electron temperature estimation’ in the revised SM. Therefore, the noise value remaining almost constant below 50mK corresponds to the ballistic

regime of upstream heat flow. This is further confirmed by nearly identical values of noise measured for two different lengths (4 μm and 10 μm) and by an excellent agreement with the theoretically calculated noise based on assumption of no thermal equilibration between counter-propagating edge modes for $\nu=2/3$ state. As is pointed out in the manuscript, the decrease of noise at $T > 50$ mK shows that one of the two mechanisms, thermal equilibration within the edge or dissipation to the environment (or their combination), becomes operative at these elevated temperatures. Each of these mechanisms would also lead to the reduction of noise as a function of length. This is in full consistency with our observation that, at the elevated temperatures, the noise for $L=10\mu\text{m}$ is lower than for $L=4\mu\text{m}$. The law of decay for both the mechanisms should be different. However, the data of two sample sizes are not enough to extract reliably the law, which is left for future research.

'5. In Figure S3, the lowest temperature in S3a is 20mK, and lowest temperature in S3b is slightly smaller than 0.2K. The other four points in S3b seem to be 1K, 775mK, 605mK and 475mK, respectively, same as those values in S3a. Are S3a and S3b from the same set of data?'

First, we confirm that Figs. S6a and S6b of the revised SM (Figs. S3a and S3b of the original version) belong to the same data set. In Fig. S6a, we have shown measured noise at zero bias current at $\nu=1$ filling factor for different bath temperatures, starting from bath temperature of 20mK to 1K. In Fig. S6b, we plot S_v/BW (at resonance frequency) versus temperature to calculate the gain and intrinsic noise of the amplifier as mentioned before. What has probably caused confusion on the part of the Reviewer is that the 20 mK curve in the panel (a) was not reflected by a data point in panel (b). This indeed was not explained in the previous version, so that the question by the Reviewer is fully justified. The reason we did not keep the $T_{\text{bath}} = 20\text{mK}$ data in Fig. S3b of the previous version is because at very low bath temperatures, electron temperature might deviate from the bath temperature. At the same time, at higher temperature such as 50mK or above, the bath temperature and electron temperature are expected to be essentially identical. It follows that this procedure minimizes an error in calculating the gain and intrinsic noise of the amplifier. We have clarified this in detail in the revised SM, see new section S6 "Gain and electron temperature estimation". Furthermore, in the revised version of Fig. S6b (former Fig. S3b) we put also a point corresponding to $T_{\text{bath}} = 20$ mK. It is seen that it is located very close to the extrapolated line, which is a manifestation of the fact that the electron temperature is in fact close to the bath temperature. We have also added Fig. S7 that illustrates a similar analysis for the $2/3$ plateau.

Reviewer # 2 (Remarks to the author):

'The manuscript from Ravi Kumar et al. reports on the experimental observation of upstream edge modes in fractional quantum Hall (FQH) states at the filling factors $\nu=2/3$ and $\nu=3/5$ in bilayer graphene. The main results are summarized below.

1. Excess upstream noise proportional to the injecting current was detected.

2. The average electric current in the upstream direction was measured to be nearly zero.

3. When the temperature is lower than 50mK, the amplitude of the detected upstream noise remains nearly identical for two different propagation lengths $L=4\mu\text{m}$ and $L=10\mu\text{m}$. By combining the above results, the authors concluded that the edges of FQH states at $\nu=2/3$ and $\nu=3/5$ possess upstream modes that only carry energy but not charges. The existence of upstream neutral modes agrees with the well-known picture of hole-conjugated FQH states. Furthermore, the upstream heat transport remains ballistic in the parameter range investigated in the experiment, which implies a lack of thermal equilibration in the sample. This feature is consistent with a recent experiment performed by the same group of authors (Ref. [15]). I think the experiment was performed systematically and all procedures are explained clearly in the manuscript. The authors went through necessary checking to eliminate alternative explanations, which allowed them to reach the above conclusions unambiguously. Also, the experimental results agree nicely with the theoretical analysis provided in the manuscript. I can easily follow the presentation (both experimental and theoretical parts) in the manuscript. On the significance of the results, I agree with the authors that this work has provided more direct evidence for the existence of upstream neutral edge modes in $\nu=2/3$ and $\nu=3/5$ FQH states in bilayer graphene. This point is clearly addressed in the main text. Based on the clear presentation, significance of the results, and high quality of the work, I would recommend a publication of the manuscript in Nature Communications.'

We thank the Reviewer for this summary of our work. We are very pleased to learn that the Reviewer finds our results significant for the field providing more direct evidence of upstream modes in bilayer graphene. Furthermore, we are delighted to read that the Reviewer praises our work for the clear presentation, significance of the results, and high quality of the work. To cap it all, we are very delighted to read that the Reviewer recommends our manuscript for publication in Nature Communications.

'To further improve the manuscript, I list my suggestions below.'

We thank the Reviewer for these comments and recommendations. We have taken all of them into account in the revised version, which has considerably improved the clarity of presentation.

Below we answer the Reviewer's comment one by one and specify the changes made in the manuscript in response to each of them.

'1. In the second paragraph of the main text, the authors cited Ref. [9] to discuss the expected two-terminal electrical conductance in non-equilibrated and equilibrated $\nu=2/3$ edges. I believe it is more respectful to cite Ref. [50] by Kane, Fisher, and Polchinski there as well.'

We fully agree that the paper by Kane, Fisher, and Polchinski (KFP) represented a major step in understanding the physics of the $2/3$ edge (and, more generally, QH edges with counterpropagating modes), paving the way to many central developments in the field. As a response to this comment by the Referee, we have included a citation of this work (which is Ref. 5 in the revised version) in the first paragraph of the Introduction.

We would like to point out, though that, while the renormalization-group analysis of disorder and interaction in the $2/3$ edge in the KFP paper is fully correct and fundamentally important, the conclusions concerning the two-terminal conductance are largely misleading. In particular, KFP stated that, in the absence of random tunneling, the two-terminal conductance depends on the inter-mode interaction strength on the edge, which is not correct: Assuming the contacts to screen interactions, the value of this conductance is universally $4/3e^2/h$, independent of the interaction. Also, KFP state that the experimentally observed value $2/3 e^2/h$ of the conductance implies a renormalization towards the strong-interaction fixed point discovered in their work. This is also not true: in fact, the conductance is $2/3e^2/h$ in the regime of strong charge equilibration, independently of the value of inter-mode interaction. While elaborating on these statements is beyond the scope of the present work, we have included appropriate references that analyzed and derived the values of the two-terminal conductance under different conditions. The upshot of all this is that we feel it would be rather confusing for the reader to cite the KFP paper in connection with the values of the two-terminal conductance. At the same time, the comment by the Reviewer has motivated us to add a reference to a slightly later paper by Kane and Fisher (Ref. 12 of the revised version), where the universality of the value $2/3e^2/h$ of the conductance in the charge-equilibrated regime was pointed out for the first time.

'2. In the same paragraph, the authors cited the two-terminal electrical conductance measurement in Ref. [10] to validate the existence of upstream modes at $\nu=2/3$. The article: "F. Lafont, A. Rosenblatt, M. Heiblum, and V. Umansky, Science 363, 54 (2019)" also reported experimental signatures of a possible upstream electric current in the spin-unpolarized $\nu=2/3$ FQH edge in a short sample. Although the two-terminal conductance $4e^2/3h$ was not directly measured there, I think it may be appropriate to cite that article as well. As a follow-up question, can the authors briefly comment whether the $\nu=2/3$ FQH state in the bilayer graphene is expected to be spin-polarized or spin-unpolarized?'

We fully agree with this suggestion of the Reviewer. We have incorporated the reference in the revised manuscript. We have added the following comment in the Introduction section of the revised manuscript in the context of a discussion of previous results on electric conductance of

GaAs edges at $\nu = 2/3$. We point out there that, for spin-unpolarized $2/3$ edge in GaAs system, the paper by Lafont et al (that has been added as Ref. 15 to the reference list) has observed at shortest lengths an increase of the conductance from $2/3$ to 0.73 (in units of e^2/h), providing an indication of an upstream mode. We also mention there that the conductance remained far from $4/3$, and that for the spin-polarized $2/3$ state no deviation from the value $2/3$ was detected.

To the follow-up question: The ground state of $2/3$ state of bilayer graphene (BLG) is much more complex compared to the GaAs-based 2DEG because of the presence of $SU(4)$ symmetry in BLG. There are several reports (see, in particular, **Science 358, 648-652 (2017)**) on the phase transition of $2/3$ state of BLG as the transverse electric field is changed. These transitions are associated with the change in the layer polarization of BLG. Existing explanations of these transitions assume that the ground state of $2/3$ phase is spin-polarized. Therefore, there exists evidence (however indirect) that the $2/3$ FQH state in bilayer graphene is most likely spin-polarized. More work is needed to answer this question with full certainty.

'3. I think Fig. 1(b) can be more informative. For example, the figure can first illustrate a downstream charge mode responsible for $\sigma_{xy} = e^2/h$ and an upstream charge mode for $\sigma_{xy} = (1/3)e^2/h$. Then, the same figure shows that these edge modes after charge equilibration become a single downstream charge mode responsible for $\sigma_{xy} = (2/3)e^2/h$ and a neutral upstream mode. By doing so, the readers (especially non experts in quantum Hall physics) can better understand the importance of charge equilibration, and the underlying reason for the upstream mode carrying energy only. I suggest the authors include a corresponding figure for the $\nu = 3/5$ FQH edge as well.'

We thank the Reviewer for this recommendation to further improve the presentation of Fig. 1b in order to better visualize different transport regimes. This will be definitely for benefit of the readers, especially non-experts in quantum Hall physics, helping to understand the impact of charge equilibration. Based on the Reviewer's suggestion, we have modified Fig. 1b in the revised manuscript. Specifically, Fig. 1b now contains three panels, two of which correspond to the two-terminal conductance in $2/3$ state in the absence and presence of charge equilibration. We have correspondingly modified and extended the caption to Fig. 1. Furthermore, following the recommendation of the Reviewer, we have added a new section S2 'Edge structure of $\nu = 3/5$ FQH state' in the revised SM. This section contains a new figure S2, where the charge-non-equilibrated and charge-equilibrated regimes of the two-terminal conductance are illustrated for the $\nu = 3/5$ state.

'4. There are several typos in the Supplementary Material which I can notice. In S2, "If one plot" should be If one plots. In S5, "remains same" should be remains the same. In S8, there is a typo "Gj ← i".'

We thank the Reviewer for pointing out typos in the manuscript. We have corrected them in the revised version.

Reviewer # 3 (Remarks to the author):

'This work studies noise measurement in bilayer graphene in the FQHE regime. The authors observed current-dependent excess noise in filling factors 2/3 and 3/5 but not in 1/3 and 1. The observations provide evidence of upstream modes in the hole-conjugate states. The manuscript is concise and clearly written.'

We are very pleased to read this positive assessment of our manuscript by the Reviewer.

'I have a few questions/comments.'

We thank the Reviewer for very useful questions and comments. We have taken into account all of them in the revised version which has allowed us to substantially improve the clarity of the paper in a number of important aspects. Below we answer the Reviewer's comment one by one and specify the changes made in the manuscript in response to each of them.

'1) In the abstract the authors described the devices studied in this work to be "dual-gated bilayer graphene". But there is no description in the manuscript or the supplementary information regarding the dual-gate, neither in terms of device fabrication nor measurement. What is the significance of dual-gate here? And is the 2DES being bilayer graphene important? What do the authors expect for monolayer graphene?'

This is an important point. The presence of the top graphite gate screens the disorder and helps to improve the quality of the sample. In the literature, it has been found that in order to observe well-developed FQH states (particularly those below filling factor 1), one needs a top graphite gate/metallic gate. We have added the discussion of this issue in a new section 'Device and electrical response' of the revised manuscript: *"For our measurements, we have used two dual-graphite gated hexagonal-boron-nitride (hBN)-encapsulated bilayer graphene devices, which are fabricated using the standard dry transfer pickup technique, see Method Section and Supplementary Material (SM), section S1. To observe at modest magnetic fields, well-developed FQH states, which are highly susceptible to disorder (below $\nu < 1$), screening layers are required. The presence of the top graphite gate serves that purpose and improve the quality of the device. The device schematic is shown in Fig. 1c. The zero-magnetic-field gate response of the device is shown in SI (S1). From this data, the mobility of the device is extracted and found to be $\sim 1.7 \times 10^5 \text{cm}^2 \text{V}^{-1} \text{s}^{-1}$ and $\sim 3 \times 10^5 \text{cm}^2 \text{V}^{-1} \text{s}^{-1}$ for the devices 1 and 2, respectively."*

Potential advantages of graphene in comparison with GaAs structures are discussed in the Introduction: *'The emergence of the two-dimensional graphene platform opened up a new era in the study of FQH physics, with different Landau level structure (as compared with traditional GaAs heterostructures), new fractions, and enriched family of quantum-Hall states due to an interplay of spin, orbital, and valley degrees of freedom. Furthermore, graphene features an unprecedented*

sharp confining potential and is thus expected to exhibit bulk-edge correspondence without additional complex edge reconstruction. Also, in view of the sharp edge, one expects strong interaction between the edge modes in graphene, which may give access to regimes that are difficult to reach in GaAs structures.' In the revised version, we have added in this context the following sentence highlighting the advantage of bilayer graphene over a single layer: *'Moreover, the presence of layer degree of freedom offers more tunability in bilayer graphene'*.

'2) From all what I can see in the manuscript and in the supplementary information, the noise measured here appears to be a voltage noise. For example, Figure S3 which plots S_V is easy to understand. Under a noiseless current, there are voltage fluctuations while the current should conserve. But in the main text the authors plot S_I . How did the authors convert to the current noise?'

We thank the Reviewer for pointing out the need to explain how we convert voltage noise to current noise. The reviewer is right that what is actually measured is the voltage noise (S_V). The excess noise due to bias current is calculated by subtracting the noise value at zero bias current from the noise at finite bias, i.e., $\delta S_V = S_V(I) - S_V(I = 0)$. To convert the excess voltage noise to the excess current noise, we divide the excess voltage noise by a square of the resistance. We recall that, in the present experiment, we have measured the voltage noise at a given integer or fractional filling where resistance of the system is precisely known, given by $R = h/\nu e^2$. The conversion formula is thus given by:

$$S_I = \frac{\delta S_V}{(h/\nu e^2)^2}$$

We note that this conversion is essentially a matter of convenience.

We have included a discussion of this issue in a new section 'Noise and analysis' of the revised manuscript and in the new section S7 "Noise analysis" of the revised Supplementary Material.

'3) There are various sources of noise (thermal noise, 1/f noise, shot noise, etc.). Since noise measurement is the technical key of this work, the authors should have a more in-depth discussion on the noise both in terms of measurements and analysis. In the current manuscript this part is rather too vague. '

We agree with the Reviewer that this aspect should have been explained in more detail. In response to this recommendation by the Reviewer, we have added an extended discussion of the noise in the first paragraph of the new section 'Noise and analysis' of the revised manuscript. We detail there first what are contributions to the equilibrium noise measured at the amplifier, Eq. (1). Further, we explain that what we study is the excess noise obtained as a difference between the noise measured at finite bias current and at zero bias current. Under this subtraction, all current-independent contributions (related to the intrinsic noise of the amplifier) cancel. We are thus left with the excess noise due to bias current, which is an intrinsic property of the FQH edge. Furthermore, as we point out in the newly added paragraph, we avoid 1/f noise contribution by

measuring the voltage noise at frequency of $\sim 763\text{kHz}$, at which $1/f$ noise contribution is totally negligible. Indeed, $1/f$ noise contribution usually becomes negligible already for frequencies above few tens of Hz, while our frequency is several orders of magnitude larger.

'4) It helps to include some basic characteristics of the devices, either in the main text or in the supplementary information. For example, the author claim "high mobility" in the main text. What is the mobility here?'

Again, we fully agree with this recommendation of the Reviewer. Following it, we present in a section "Device and electrical response" of the revised manuscript basic characteristics of the device. In particular, we quote there now the value of the mobility for our devices: $\sim 1.7 \times 10^5 \text{cm}^2 \text{V}^{-1} \text{s}^{-1}$ and $\sim 3 \times 10^5 \text{cm}^2 \text{V}^{-1} \text{s}^{-1}$ for device-1 and device-2, respectively. Details of the analysis of the mobility are presented in Section S1 "Device fabrication and characteristics", which has been substantially extended in the revised version. Specifically, see the added text in the end of that section [around Eq. (S1)] as well Fig. S1 that was strongly extended in comparison with the original version: panels c and d added, with a detailed caption.

REVIEWERS' COMMENTS

Reviewer #1 (Remarks to the Author):

The authors have responded to my questions with convincing experimental data and analysis. Although some of the problems needed to be addressed with future research, it is natural to have minor puzzling issues in an experiment. The limitation of the device, such as non-fully developed $1/3$ state, should not cover the useful information from the $2/3$ state. I appreciate the authors' patience and the additional details, and I support the publication of this manuscript.

Reviewer #2 (Remarks to the Author):

The authors have addressed all my concerns and incorporated my suggestions into the revised manuscript. I find the manuscript more comprehensive and contains sufficient background information for non-experts in quantum Hall physics. Therefore, I recommend this article to be published in the present form.

Reviewer #3 (Remarks to the Author):

The authors' reply is satisfactory. I'm ok with accepting the manuscript at its current form.